# RawEnhancer: Attentive Multi-Exposure Selection for Bracketing Image Reconstruction

## Abstract

Exposure bracketing is a crucial technique for capturing high-quality images in real-world scenes. The key challenge lies in effectively fusing complementary information across bracketing exposures while considering the unique characteristics of each exposure. To address this, we develop an attentive multi-exposure selection network, dubbed RawEnhancer, built on an asymmetric encoder-decoder architecture for visual-pleasing bracketing image reconstruction. Specifically, we first present an exposure-aware selective attention (EASA) that leverages gradient magnitude and color channel intensity priors from the RAW sensor data to achieve a finer representation. To enable the model to handle multiple exposures, we construct a weight-sharing encoder with the EASA layers to efficiently encode multi-scale information of each individual exposure. These multi-scale representations are then progressively fused in the decoder part through an iterative feature selection (IFS) strategy, which adaptively integrates complementary information across multiple exposures at each scale. Experimental results show that the proposed RawEnhancer performs favorably against state-of-the-art ones on benchmark datasets in terms of accuracy and model efficiency.

## 1 Introduction

Bracketing image reconstruction is a critical computational photography technique that overcomes the dynamic range limitations of single-shot by fusing image sequences captured under varying exposure conditions. This technique, which enables the production of high-quality high dynamic range (HDR) images, holds significant importance for enhancing imaging quality. However, it is highly challenging due to exposure variations and misalignments caused by scene or camera motion.

Over the past decade, bracketing image reconstruction Zhang et al. (2025); Lee et al. (2025); Chi et al. (2023); Chen et al. (2025) has witnessed significant performance improvements, driven by the rapid development of deep learning techniques and large-scale training data, leading to a surge of innovative solutions Dudhane et al. (2023; 2022); Zhang et al. (2025); Liu et al. (2022); Kalantari & Ramamoorthi (2017); Prabhakar et al. (2019); Liu et al. (2023; 2021b); Yan et al. (2019b); Wei et al. (2023); Zhang et al. (2022). Among these, sRGB-domain multi-exposure HDR reconstruction methods Kalantari & Ramamoorthi (2017); Prabhakar et al. (2019); Yan et al. (2019a); Tel et al. (2023); Wang et al. (2023a); Liu et al. (2022); Kong et al. (2024); Yan et al. (2019b; 2020) are particularly prominent. Researchers have continuously optimized network architectures, from early convolutional neural networks Kalantari & Ramamoorthi (2017); Prabhakar et al. (2019); Yan et al. (2019b) to more recent Transformer-based models Liu et al. (2022); Tel et al. (2023); Wang et al. (2023a), which have significantly boosted reconstruction performance. However, these methods are typically trained on idealized datasets featuring simplified degradations and clean scene content. This limitation makes them struggle with the complex degradations found in real-world shooting conditions, thus restricting their practical application. Furthermore, sRGB images, having undergone an in-camera ISP pipeline, suffer from significant loss of RAW sensor information, which restricts the potential for high-fidelity restoration. Thus, incorporating more realistic degradations and exploring restoration in the RAW domain are essential for practical HDR imaging.

To mitigate these challenges, there is a growing trend to explore multi-exposure image fusion using noisy, blurry, or misaligned frames with varying exposures Chi et al. (2023); Liu et al. (2023); Lee et al. (2025); Zhang et al. (2022; 2025); Guo et al. (2022). This has been driven by the emergence of

new RAW benchmarks that better simulate real-world degradations. For instance, the MobileHDR dataset Liu et al. (2023) includes diverse daytime and nighttime scenes under varying noise levels; the RawFusion dataset Lee et al. (2025) features more realistic noise simulation and more complex degradation patterns. Additionally, the BracketIRE dataset Zhang et al. (2025) considers most multi-exposure factors, unifying denoising, deblurring, and HDR reconstruction into a single bracketing image restoration and enhancement task.

Building upon these datasets, researchers have proposed various methods for leveraging multi-exposure information. Joint-HDRDN Liu et al. (2023) uses cross-attention for feature alignment; TMRNet Zhang et al. (2025) adopts a progressive feature propagation mechanism with optical flow; and RASD Qiu et al. (2025) proposes a temporal-spatial decoupling strategy that separates alignment from the restoration task. While these methods achieve good reconstruction results, they generally align multi-exposure inputs at an early stage of the model before subsequent processing. This manner relies on accurate alignment. However, the performance of alignment driven by optical flow Kong et al. (2024); Kalantari & Ramamoorthi (2017); Prabhakar et al. (2019); Zhang et al. (2025) or attention mechanisms Yan et al. (2019a); Liu et al. (2021b; 2022; 2023); Tel et al. (2023) degrades when exposure differs significantly, as the luminance consistency assumption breaks down in overexposed or dark regions, leading to unreliable optical flow fields and distorted attention maps. Additionally, existing methods primarily focus on exploiting information across frames, overlooking the intrinsic prior information contained within each exposure itself. Therefore, effectively exploring correlations among multiple exposures while fully leveraging the inherent priors within each RAW image is vital for high-quality bracketing image reconstruction.

In this paper, we present an attentive multi-exposure selection network for generating high-quality HDR image. Unlike previous methods that directly merge multi-exposure images, our approach first encodes the multi-scale information of each individual exposure and then performs multi-exposure fusion in the decoder using an asymmetric encoder-decoder architecture. Specifically, we introduce an exposure-aware selective attention (EASA) layer that leverages the gradient magnitude and color channel intensity prior of RAW sensor data to achieve a finer representation. To enable the model to handle multiple exposures, we construct a weight-sharing encoder with EASA layers. To facilitate comprehensive interactions among multiple exposures, we further introduce an iterative feature selection (IFS) module that adaptively aggregates complementary information across bracketing inputs, eliminating the need for explicit feature alignment. Extensive experimental results show that our proposed method is capable of generating high-quality results under various exposure conditions.

Our contributions can be summarized as follows:

- We propose an exposure-aware selective attention (EASA) layer to compose the encoder, which is used to selectively encode multi-scale contextual information for each individual.

- We propose an iterative feature selection (IFS) module that effectively fuses valuable features across exposure sequences without requiring alignment.

- We conduct quantitative and qualitative evaluations for our proposed method on different benchmarks, and experimental results demonstrate that our RawEnhancer achieves favorable reconstruction performance.

## 2 RELATED WORK

**Burst Image Restoration.** Burst image restoration aims to reconstruct a single high-quality image by fusing multiple consecutively captured frames, and is widely used for tasks such as denoising Dudhane et al. (2022); Godard et al. (2018); Mildenhall et al. (2018), deblurring Wieschollek et al. (2017); Peña et al. (2019), and super-resolution Wei et al. (2023); Dudhane et al. (2023); Di et al. (2025); Bhat et al. (2021); Liu et al. (2025); Wu et al. (2023). The core of this research revolves around inter-frame alignment and feature fusion. Methods for inter-frame alignment can be broadly categorized into explicit alignment and implicit alignment. Explicit alignment uses dedicated modules to estimate motion between frames, employing techniques like optical flow Ranjan & Black (2017); Bhat et al. (2021) and deformable convolution Dai et al. (2017); Wu et al. (2023); Wei et al. (2023) to dynamically compensate for pixel shifts. Implicit alignment uses attention mechanisms Dudhane et al. (2023; 2022) to learn inter-frame correlations and adaptively fuse features. For the feature fusion, various strategies are used to merge the aligned multi-frame information. These include weighted-based

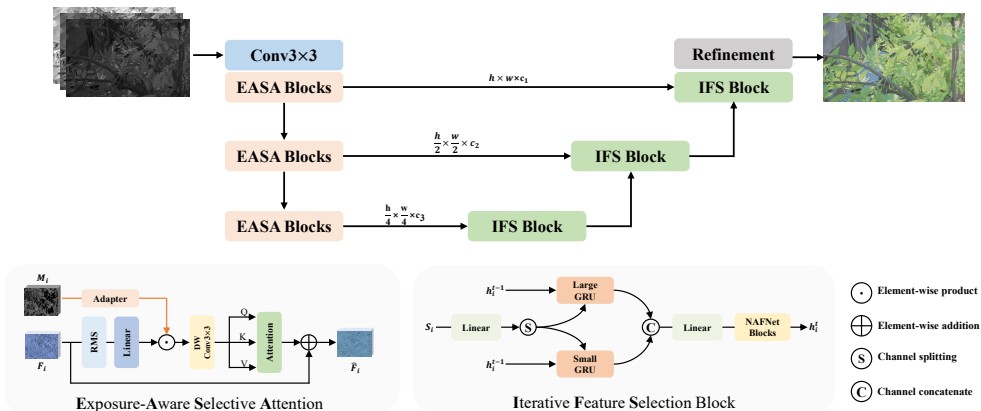

Figure 1: **Network architecture of the proposed RawEnhancer**. For the encoder module, we introduce multiple exposure-aware selective attention (EASA) blocks to encode finer multi-scale representations. For the decoder module, we develop an iterative feature selection block that dynamically merges useful information from multiple exposures without requiring alignment.

mechanisms Bhat et al. (2021), kernel prediction Mildenhall et al. (2018), cross-attention Mehta et al. (2023), and recursive fusion Guo et al. (2022). Although these methods can achieve impressive results in scenes with a fixed exposure, a key limitation is that the reconstructed images often have a limited dynamic range.

**Multi-Exposure HDR Imaging.** Exposure bracketing is widely used for HDR image reconstruction. Traditional methods rely on specific priors Lee et al. (2014); Sen et al. (2012); Khan et al. (2006) for preprocessing, but they perform poorly in dynamic scenarios. Recently, deep learning-based methods Kalantari & Ramamoorthi (2017); Yan et al. (2021); Prabhakar et al. (2019); Tel et al. (2023); Niu et al. (2021); Liu et al. (2022; 2021b); Kong et al. (2024); Chen et al. (2025) have made significant progress. Optical flow-based methods Kalantari & Ramamoorthi (2017); Yan et al. (2019b) align low dynamic range (LDR) images by estimating motion, but they struggle with exposure variations. In contrast, Transformer-based approaches Liu et al. (2022); Tel et al. (2023); Wang et al. (2023a) use self-attention or its variants to capture long-range dependencies and adaptively fuse features for better multi-exposure HDR imaging. Despite their good performance, these methods often rely on idealized data that lacks the realism of practical capture conditions. To address this, recent approaches Chi et al. (2023); Liu et al. (2023); Lee et al. (2025); Zhang et al. (2022; 2025); Guo et al. (2022) incorporate practical factors to improve generalization. For example, SV-HDR Chi et al. (2023) simultaneously denoises the image and blends multiple exposures into a single HDR image. Joint-HDRDN Liu et al. (2023) constructs a real-world mobile HDR dataset and uses cross-attention for effective denoising and fusion. TMRNet Zhang et al. (2025) proposes a synthetic dataset with real-world degradations and uses a progressive fusion module for feature alignment. While these methods show promising results, there is still room for improvement in model design and the utilization of scene priors.

## 3 PROPOSED METHOD

Figure 1 illustrates the overall architecture of our proposed RawEnhancer, which adopts an asymmetric encoder-decoder framework to draw information from all bracketing frames for high-quality HDR reconstruction. Unlike previous methods that first perform multi-frame merging via optical flow Kong et al. (2024); Kalantari & Ramamoorthi (2017); Prabhakar et al. (2019) or attention mechanisms Wang et al. (2023a); Tel et al. (2023) followed by image reconstruction process, we introduce an "encoding-then-fusion" format to fully leverage the inherent priors and complementary information within multi-exposure images. This manner could achieve efficient and effective exposure fusion without requiring any alignment operations.

Given a set of low-quality LDR images $\{L_1, L_2, ..., L_t\}$ captured at different exposure levels, following the regular preprocessing pipeline of existing HDR reconstruction methods Liu et al. (2021b); Zhang et al. (2025); Liu et al. (2023), we start with the normalization and gamma correction based on the exposure time of each image. Subsequently, the contrast-enhanced images are concatenated with

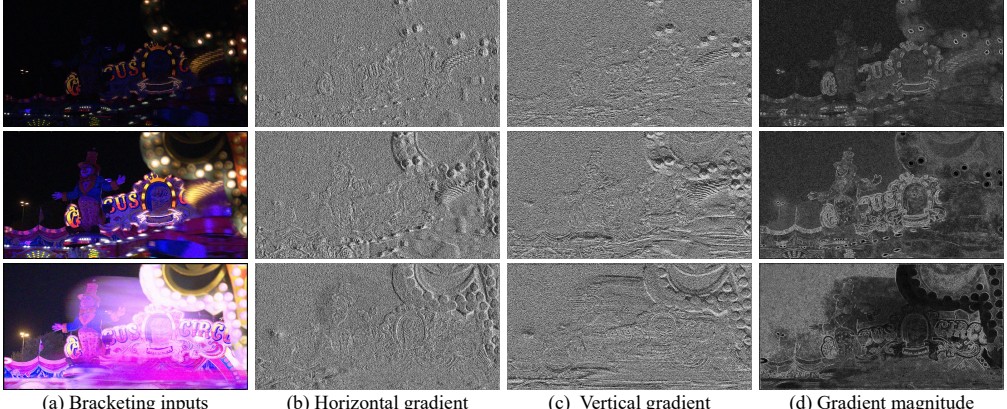

(a) Bracketing inputs      (b) Horizontal gradient      (c) Vertical gradient      (d) Gradient magnitude

Figure 2: Visualization of gradients and gradient magnitudes under different exposure inputs, where gradient magnitude consistently captures structural and edge details across varying exposures.

the original one as the input to the model. A $3 \times 3$ convolutional layer is first applied to extract the shallow features from the input $L_t$. The shallow features are then encoded with an encoder module to extract multi-scale representation, where the feature encoding is achieved by the exposure-aware selective attention (EASA) blocks. To enhance the model's ability to handle different exposures, we employ a weight-sharing encoder to process all the exposure cases. Following the encoder module, these encoded multi-scale features are fed into the decoder to generate finer representation through an iterative feature selection (IFS) strategy. Within the decoder module, two parallel GRU sub-modules with different kernel sizes adaptively integrate semantic and detail information from multi-exposure images across all scales. Finally, the merged features are used to produce high-quality HDR output through a feature refinement module.

### 3.1 EXPOSURE-AWARE SELECTIVE ATTENTION

For bracketing images, the captured content often contains overexposed or underexposed regions due to the limited dynamic range of the imaging sensor. Pixels in these regions tend to be saturated or dominated by noise, resulting in a loss of useful visual information. During the feature extraction or fusion stages, if these regions are not properly handled, the fused results are easily compromised by these degraded contents Chi et al. (2023); Hu et al. (2013); Liu et al. (2023), significantly diminishing the quality of the final reconstructed image. Therefore, effectively identifying and suppressing the impact of these invalid regions is essential for enhancing the performance of bracketing image reconstruction.

Furthermore, most existing multi-exposure HDR imaging methods employ self- or cross-attention mechanisms Wang et al. (2023a); Liu et al. (2022; 2023); Tel et al. (2023) to dynamically aggregate similar features across exposures. However, these attention mechanisms are sensitive to luminance variations, often leading to inaccurate or unreliable attention weights in overexposed or underexposed areas. This negatively impacts the utilization of useful information from well-exposed regions during global feature aggregation.

To alleviate the aforementioned challenges, we present an exposure-aware selective attention module that identifies valid regions and dynamically aggregates features, enhancing the recovery of clipped pixels. Specifically, given the LDR input $L_t$ and its corresponding encoded features $F_i$ at scale $i$, we first compute the gradient magnitude $M_i$ of $L_t$. This magnitude is then used to highlight the informative features within $F_i$ via an element-wise product manner. This operation is formulated as:

$$
\begin{aligned}
\nabla L_t &= \left( \frac{\partial L_t}{\partial x}, \frac{\partial L_t}{\partial y} \right), \\
M_i &= \sqrt{ \left( \frac{\partial L_t}{\partial x} \right)^2 + \left( \frac{\partial L_t}{\partial y} \right)^2 }, \\
\hat{F}_i &= Conv_{1 \times 1}(RMSNorm(F_i)) \odot \mathcal{C}_{ada}(M_i),
\end{aligned}
\tag{1}
$$

where $\left(\frac{\partial L_t}{\partial x}, \frac{\partial L_t}{\partial y}\right)$ denotes the gradient of the image $L_t$ at the point $(x, y)$, $M_i$ represents the corresponding gradient magnitude, and $\hat{F}_i$ is the gated output. Due to a channel dimension mismatch between the gradient magnitude $M_i$ and the feature maps $F_i$, we introduce an adapter layer $\mathcal{C}_{ada}$, composed of two linear layers, to project $M_i$ into a compatible channel space for subsequent aggregation. Additionally, we apply root mean square normalization $RMSNorm(\cdot)$ Zhang & Sennrich (2019) to the input features to improve training stability.

Although gradients offer a direct way to detect content-rich regions, their reliability is reduced by noise and blur in complex exposure conditions, as shown in Figure 2. Thus, we leverage the image gradient magnitude as a robust gating criterion.

After obtaining the gated feature $\hat{F}_i$, we need to aggregate all these useful information to form a discriminative representation. Considering the green channel prior Zou et al. (2023); Guo et al. (2021); Liu et al. (2020) and the linear intensity mapping inherent in RAW sensor data, we propose a channel-wise selective attention mechanism to leverage complementary information across channels to effectively preserve informative features under diverse exposure conditions.

Similar to Zamir et al. (2022), we first extract local features using a depth-wise convolution with kernel size of $3\times3$, then project these features into the query vector $\mathbf{Q} \in \mathbb{R}^{HW \times C}$, the key vector $\mathbf{K} \in \mathbb{R}^{HW \times C}$, and the value vector $\mathbf{V} \in \mathbb{R}^{HW \times C}$. The detailed procedure can be denoted as:

$$
\begin{aligned}
\mathbf{Q}, \mathbf{K}, \mathbf{V} &= \mathcal{S}(DWConv_{3\times3}(\hat{F}_i)), \\
\mathbf{A} &= \text{ReLU}\big(\mathbf{Q} \cdot \mathbf{K}^\top / \alpha\big), \\
\widetilde{F}_i &= (\mathbf{A} \cdot \mathbf{V}) + \hat{F}_i,
\end{aligned}
\tag{2}
$$

where $\mathcal{S}(\cdot)$ denotes the channel splitting operation, $\alpha$ represents the scaling factor, and $\widetilde{F}_i$ refers to the processed feature. Subsequently, $\widetilde{F}_i$ is normalized by the RMSNorm operator Zhang & Sennrich (2019) and further refined through a gated feed-forward network Zamir et al. (2022). To enhance the selection capacity of representative features under varying exposures, we choose ReLU to generate attention maps that increase sparsity while filtering out irrelevant information Li et al. (2023).

During the multi-scale feature encoding process, we employ strided convolution for feature downsampling, enabling the network to learn more reliable hierarchical representations. For the calculation of gradient magnitude at each scale, we downsample the input image using PixelUnshuffle layer Shi et al. (2016) before calculating the magnitude. Compared to directly applying average pooling to the input, this strategy better preserves structural information, allowing for a more accurate estimation of the gradient magnitude.

## 3.2 ITERATIVE FEATURE SELECTION

Merging valuable information across multiple exposures is a crucial step toward generating high-quality HDR images. Unlike conventional approaches that rely on a pre-defined reference image, our proposed method operates in a reference-free manner, enabling more comprehensive utilization of informative content from multi-exposure inputs. This design eliminates the need for explicit feature alignment, thus avoiding the inaccuracies associated with optical flow-based or correlation-based methods Zhang et al. (2025); Kong et al. (2024); Kalantari & Ramamoorthi (2017); Liu et al. (2022; 2023). Instead, we adopt an iterative fusion selection strategy that progressively integrates and refines the representative features. To keep spatial consistency, we jointly model both local and non-local contexts at each scale, enabling dynamic and context-aware feature aggregation.

Given the $i$-th scale feature set $S_i$ consisting of encoded features $\widetilde{F}_i$ from different exposure images, we perform iterative feature fusion over $S_i$. Specifically, at iteration $t$, the current-scale input features $s_i$ are first mapped via a linear layer to obtain two separate features $s_i^1$ and $s_i^2$. These, together with the fused state of the previous iteration $h_i^{t-1}$, are fed into a recurrent fusion module. This module consists of two parallel GRU networks with different kernel sizes, which are used to capture short- and long-range feature dependencies across multiple exposures, respectively. As the original convolution with a large kernel size is parameter-heavy, we decompose it into a $1 \times 1$ convolution and a $3 \times 3$ depth-wise convolution. Their outputs are then concatenated and fused by a $1 \times 1$ convolution to form the intermediate fused feature $z_t$ ,which is subsequently optimized through several NAFNet

Table 1: Quantitative comparison with state-of-the-art methods on the BracketIRE-syn Zhang et al. (2025), MobileHDR Liu et al. (2023), and RawFusion Lee et al. (2025) datasets.

| Methods | BracketIRE-syn | MobileHDR | | RawFusion |
|---|---|---|---|---|
| | PSNR/SSIM/LPIPS↓ | PSNR/SSIM (RAW, $\mu$) | PSNR/SSIM (RAW, $l$) | NIQE↓/CLIPIQA↑/MUSIQ↑ |
| BIPNet | 36.92/0.9331/0.148 | 38.74/0.9606 | 35.64/0.9895 | 5.960/0.4555/59.97 |
| Burstormer | 37.06/0.9344/0.151 | 38.64/0.9586 | 36.28/0.9897 | **5.868**/0.4569/59.46 |
| FBANet | 36.85/0.9250/0.148 | 37.99/0.9530 | 35.81/0.9888 | 6.147/0.4434/60.73 |
| HDR-Transformer | 37.62/0.9356/0.129 | 38.54/0.9599 | 35.42/0.9873 | 6.082/0.4410/60.60 |
| SAFNet | - | 37.57/0.9538 | 35.54/0.9886 | 6.097/0.4578/59.49 |
| Joint-HDRDN | - | 38.92/**0.9616** | 36.37/0.9888 | 6.098/0.4374/60.80 |
| TMRNet | **39.35**/0.9516/**0.112** | 38.13/0.9550 | 35.97/0.9895 | 6.249/0.4462/61.24 |
| RawEnhancer (Ours) | 39.29/**0.9524/0.112** | **39.19**/0.9613 | **37.05/0.9908** | 6.073/**0.4607/61.42** |

blocks Chen et al. (2022). This process can be described as:

$$
\begin{aligned}
s_i^1, s_i^2 &= \mathcal{S}(Conv_{1\times1}(s_i)), \\
h_{\text{small}}^t &= \text{GRU}_{3\times3}(h_i^{t-1}, s_i^1), \\
h_{\text{large}}^t &= \text{GRU}_{7\times7}(h_i^{t-1}, s_i^2), \\
z^t &= Conv_{1\times1}([h_{\text{small}}^t, h_{\text{large}}^t]), \\
h_i^t &= \mathcal{N}_f(z^t),
\end{aligned}
\tag{3}
$$

where $[\cdot]$ represents a channel concatenation operation, $\mathcal{N}_f(\cdot)$ is a feature refinement network composed of NAFNet blocks.

As the iterative process progresses, the model progressively integrates and strengthens informative features from multiple exposure inputs. The updated fused features are subsequently acted as guidance for the fusion at the next scale, enabling effective cross-scale propagation and progressive refinement.

## 4 EXPERIMENTAL RESULTS

### 4.1 DATASETS AND IMPLEMENTATION

**Datasets.** We conduct experiments on diverse RAW datasets, including BracketIRE-syn Zhang et al. (2025), MobileHDR Liu et al. (2023), and RawFusion Lee et al. (2025).

**Implementation details.** During training, we randomly crop 16 image patches of size $128 \times 128$ from LDR sequences as input. We also perform data augmentation on the input data, including random horizontal flipping and rotation. The proposed method is optimized using the Adam optimizer Kingma & Ba (2015) with $\beta_1 = 0.9$, $\beta_2 = 0.99$, and $\epsilon = 10^{-8}$. The total number of training iterations is set to 500,000. The initial learning rate is $3 \times 10^{-4}$ and is then decayed to a minimum of $1 \times 10^{-6}$ using a cosine annealing schedule Loshchilov & Hutter (2017).

Our method uses a combined loss function during the training phase, which includes L1 loss and an FFT-based frequency loss Cho et al. (2021); Sun et al. (2022; 2023), with the weights $\{1, 0.1\}$. Consistent with previous methods Zhang et al. (2025); Liu et al. (2023); Wang et al. (2023a), output images used for loss computation are $\mu$-law mapped.

Additionally, hyperparameters for the method are set as follows: channel dimensions across scales are [64, 96, 128], the number of blocks per scale are [4, 6, 8], the feature iterative fusion process runs for 3 iterations, attention heads in the EASA module are [2, 3, 4], and the expansion factor of the feed-forward network is 2. All experiments are implemented with PyTorch framework on NVIDIA GeForce RTX 3090 GPUs. Due to the page limit, we include more experimental results in the supplementary material. The training code and pre-trained models will be available to the public.

### 4.2 COMPARISONS WITH STATE-OF-THE-ART METHODS

To comprehensively evaluate the performance of the proposed method, we compare it with several representative bracketing image reconstruction approaches, including BIPNet Dudhane et al. (2022),

Table 2: Efficiency comparison with representative methods. "OOM" indicates out Of memory during inference, and "/" means that even with a patch tile strategy, the model still cannot complete the inference successfully. MUSIQ is tested on the RawFusion Lee et al. (2025) dataset.

| Methods | #Params (M) | #FLOPs (G) | #GPU Mem. (M) | Runtime (ms) | MUSIQ↑ |
|---|---|---|---|---|---|
| BIPNet | 6.28 | 6101.14 | OOM | / | 59.98 |
| Burstormer | 3.11 | 626.78 | OOM | 502.49 | 59.46 |
| HDR-Transformer | **1.69** | 117.13 | 299.04 | 125.04 | 60.60 |
| Joint-HDRDN | 1.98 | **111.67** | **228.18** | 103.27 | 60.80 |
| TMRNet | 13.29 | 1348.25 | 788.49 | 86.29 | 61.24 |
| RawEnhancer (Ours) | 4.96 | 238.76 | 241.02 | **72.79** | **61.42** |

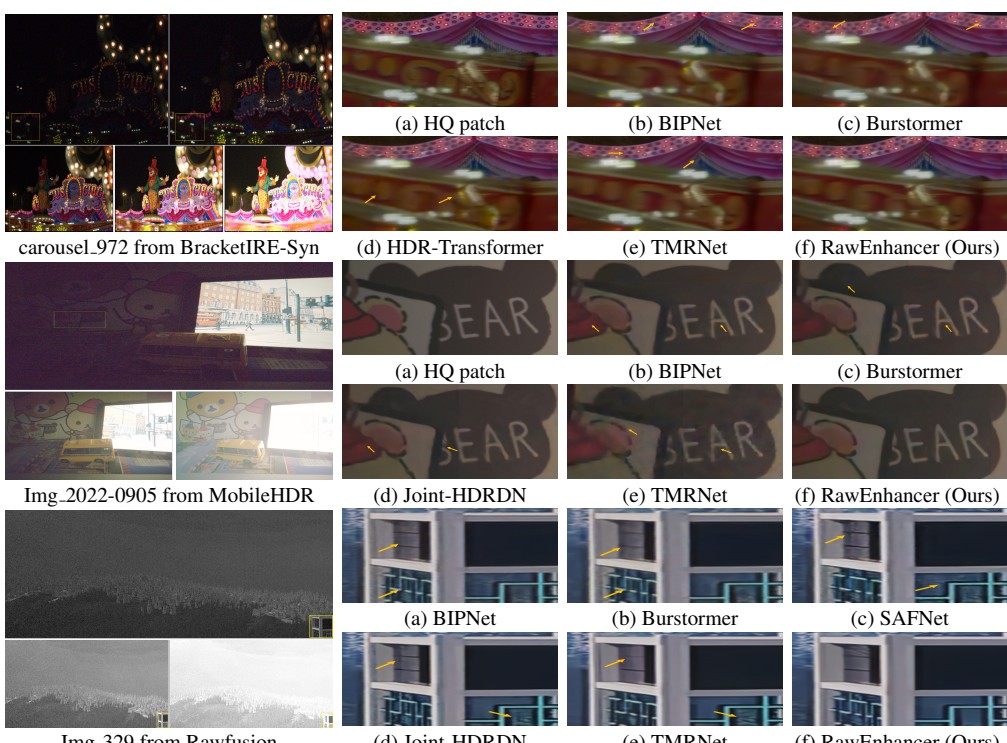

Figure 3: Visual comparisons on different scenes. The reconstructed results of the listed methods exhibit noticeable artifacts, whereas our RawEnhancer produces sharper and cleaner images.

Burstormer Dudhane et al. (2023), FBANet Wei et al. (2023), HDR-Transformer Liu et al. (2022), SAFNet Kong et al. (2024), Joint-HDRDN Liu et al. (2023), and TMRNet Zhang et al. (2025).

Table1 presents the quantitative comparison results on benchmark datasets. All listed methods are retrained on the corresponding datasets to ensure a fair comparison. SAFNet Kong et al. (2024) and Joint-HDRDN Liu et al. (2023) are excluded from evaluation on the BracketIRE-syn Zhang et al. (2025) dataset, as they only support three-exposure images as input. For evaluation metrics, the BracketIRE-syn and Mobile-HDR datasets adopted the same criteria as their original papers. Since the official online testing platform for the RawFusion dataset is no longer available, we employ commonly used no-reference image quality assessment metrics, including NIQE Mittal et al. (2013), CLIPIQA Wang et al. (2023b), and MUSIQ Ke et al. (2021), to objectively evaluate the reconstruction quality. Furthermore, to analyze model efficiency, we compare representative methods in terms of the number of parameters (#Params), #FLOPs, maximum GPU memory consumption (#GPU Mem.), and runtime, as shown in Table 2. #FLOPs are evaluated on an input of size $5 \times 4 \times 256 \times 256$. The inference time and #GPU Mem. of all models are tested with an input of size $5 \times 4 \times 512 \times 512$.
**Quantitative comparison.** Table 1 demonstrates that our RawEnhancer achieves better performance on almost test datasets. Benefiting from the EASA modules and IFS blocks, our RawEnhancer enables high-quality HDR reconstruction without explicit alignment. For instance, compared to

Table 3: Effectiveness of the EASA module.

|      | w/o EASA | WSA    | w/o Magnitude | w/ Gradient | Ours       |
|------|----------|--------|---------------|-------------|------------|
| PSNR | 36.69    | 36.99  | 37.48         | 37.56       | **37.63**  |
| SSIM | 0.9308   | 0.9345 | 0.9377        | 0.9392      | **0.9403** |

(a) Medium exposure    (b) HQ patch    (c) Window-based SA    (d) w/o EASA

(e) Long exposure    (f) w/o Magnitude    (g) w/ Gradient    (h) Ours

Figure 4: Effectiveness of the EASA module. Utilizing this module to extract features enables the model to yield better reconstruction results.

TMRNet Zhang et al. (2025), which relies on optical flow for feature alignment, our RawEnhancer achieves comparable reconstruction performance on the BracketIRE-syn Zhang et al. (2025) dataset with only 17.7% of its FLOPs. Compared with correlation-guided approaches Liu et al. (2023; 2022), our RawEnhancer has a significant PSNR gain, which is an average of 0.47dB and 1.16dB higher than Joint-HDRDN Liu et al. (2023) and HDR-Transformer Liu et al. (2022) on the Mobile-HDR Liu et al. (2023) dataset, respectively. These results validate the effectiveness of the proposed method in handling various degradations present in multi-exposure inputs.

**Efficiency comparison.** Table 2 compares the model complexity and inference cost, demonstrating that our RawEnhancer achieves a favorable trade-off between reconstruction performance and model efficiency. Compared to TMRNet Zhang et al. (2025), despite employing an iterative fusion strategy, the proposed method reduces maximum memory consumption by half, uses only 37.3% of its parameters, and achieves faster inference.

**Qualitative comparison.** Figure 3 presents visual comparisons between our proposed method and the listed methods Dudhane et al. (2022; 2023); Liu et al. (2022); Zhang et al. (2025); Liu et al. (2023); Kong et al. (2024). Compared to methods that rely on a pre-defined reference frame, our method demonstrates robust performance in diverse exposure and motion scenarios. In the first example, TMRNet Zhang et al. (2025) uses the short-exposure image as the reference, which lacks sufficient structural information in extremely dark regions, resulting in considerable detail loss in the reconstructed image (see subfigure (e) in the first row). In the second example, Joint-HDRDN Liu et al. (2023) uses the mid-exposure image as the reference and performs correlation-guided feature alignment in the shallow feature space. However, uneven exposures disrupt the attention estimation, leading to incomplete object structures and noticeable noise in the final output (see subfigure (d) in the second row).

## 5 ANALYSIS AND DISCUSSIONS

In this section, we conduct extensive ablation studies to analyze and evaluate the effectiveness of each component in the proposed RawEnhancer. All experiments are performed on the BracketIRE-syn Zhang et al. (2025) dataset. To enable a quick evaluation, we randomly selected 50 images from all four scenarios in the BracketIRE-syn testset to form the validation data. The training batch size is set to 8, the total number of training iterations is 300,000, and all other settings remain unchanged.

**Effectiveness of the EASA module.** The proposed EASA module dynamically selects informative regions from multi-exposure inputs. To evaluate its effectiveness and necessity, we first conduct an ablation by removing this module. Without EASA, PSNR drops by 0.94dB on the validation set, demonstrating its importance. Next, we replace EASA with a window-based self-attention module Liu et al. (2021a; 2023; 2022) (WSA for short). This substitution leads to a significant performance

Table 4: Effectiveness of the IFS block.

|  | PFP | w/ only GRU$_{3\times3}$ | w/ only GRU$_{7\times7}$ | Iteration 1 | Iteration 2 | Ours |
|---|---|---|---|---|---|---|
| PSNR | 37.48 | 36.75 | 36.73 | 37.51 | 37.58 | **37.63** |
| SSIM | 0.9398 | 0.9316 | 0.9307 | 0.9392 | 0.9391 | **0.9403** |

(a) Medium exposure     (b) HQ patch     (c) PFP     (d) w/ only GRU$_{3\times3}$

(e) Long exposure     (f) w/ only GRU$_{7\times7}$     (g) Iteration 2     (h) Ours

Figure 5: Effectiveness of the IFS block. The proposed IFS block can reconstruct better results.

drop of 0.64dB. As discussed in Section 3.1, self- or cross-attention mechanisms are susceptible to disturbance from extremely bright or dark regions, which negatively affects the final output. We further verify the necessity of incorporating gradient magnitude. When gradient magnitude is removed (w/o Magnitude for short), both PSNR and SSIM decrease noticeably, suggesting that selective attention alone cannot fully exploit informative content across different exposures. Moreover, compared to modulating features with gradients (w/ Gradient for short), using gradient magnitude yields better results, with a clear improvement in SSIM. This result indicates that gradient magnitude provides a more reliable cue for detecting informative regions. The visual comparison in Figure 4 further proves that using the EASA module helps improve HDR reconstruction.

**Effectiveness of the IFS block.** The proposed IFS module dynamically merges informative content from multiple exposures through progressive cross-scale feature propagation, enabling high-quality HDR reconstruction. To ensure spatial consistency, both local and non-local feature interactions are performed simultaneously across multiple scales. To evaluate its effectiveness, we first compare our method to a progressive feature propagation approach Zhang et al. (2025) (PFP for short), where the first frame serves as the initial result, and information from subsequent frames is aggregated step by step. This comparison reveals a significant performance drop. Next, we investigate the impact of extracting both local and non-local information at each scale. Table 4 demonstrates that using only a single-branch GRU with either a small or large kernel size leads to a significant decrease in reconstruction performance, with PSNR dropping by 0.88 dB and 0.90 dB, respectively. Figure 5 shows that combining two GRUs with different kernel sizes allows the model to more effectively integrate complementary information from various exposures, thereby producing more accurate structures. Finally, we verify the effect of the number of iterations. As shown in Table 4, increasing the number of iterations improves both PSNR and SSIM values. To achieve a better balance between performance and efficiency, we ultimately chose 3 iterations as the default configuration.

## 6 CONCLUSION

We have presented an attentive multi-exposure selection model for efficient bracketing image restoration. We develop an effective exposure-aware selective attention (EASA) module to selectively encode multi-scale contextual information for each individual exposure. To enable the model to handle multiple exposures, we construct a weight-sharing encoder with EASA layers. To facilitate comprehensive interactions among multiple exposures, we propose an iterative feature selection (IFS) module that effectively fuses valuable features across bracketing sequences without requiring alignment. By training the proposed approach in an end-to-end manner, we show that our method performs favorably against state-of-the-art models in terms of accuracy and efficiency.

**LLM usage:** in this paper, we use LLMs to assist with grammar proofreading and polishing writing. All suggestions were carefully revised by the authors, who bear full responsibility for the final content.

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
