# RawEnhancer: Attentive Multi-Exposure Selection for Bracketing Image Reconstruction
## - Supplemental Material -

In this document, we first describe the used datasets in Section 1 and show the efficiency comparison across different iterations in Section 2 . To provide an intuitive understanding of our proposed method, we present visualizations of the intermediate features of the EASA and IFS modules in Sections 3 and 4, respectively. Finally, we provide more visual comparison results in Section 5 and demonstrate the real-world comparisons in Section 6.

## 1 DATASET DESCRIPTION

We conduct experiments on Raw datasets from diverse scenarios, including BracketIRE-syn Zhang et al. (2025), MobileHDR Liu et al. (2023), and RawFusion Lee et al. (2025).

The BracketIRE-syn dataset Zhang et al. (2025) is synthetically generated from high-quality HDR video, simulating typical degradations such as noise, blur, uneven exposure, and dynamic range compression. It comprises 1,335 image pairs, each scene containing five distinct exposure levels. A total of 1,045 pairs are assigned for training, while the remaining 290 are reserved for testing.

The MobileHDR dataset Liu et al. (2023) is captured using three types of mobile phone in both daytime and nighttime scenes, covering a wide range of noise levels from ISO 100 to 6400. This dataset contains 251 samples, each containing three exposures. Among these, 223 sets are used for training, while the remaining 28 are used for testing.

The RawFusion dataset Lee et al. (2025) is generated via a virtual pipeline, introducing real-world degradations like noise, rotation, and motion blur. Each scene includes three exposure levels, resulting in a total of 340 image pairs with a resolution of $768 \times 1536$. Among these, 300 pairs are used for training and 40 pairs are reserved for testing.

## 2 EFFICIENCY COMPARISON ACROSS DIFFERENT ITERATIONS

Table 1: efficiency comparison across different iterations.

| Iterations | PSNR | SSIM | FLOPs (G) | Runtime (ms) |
|---|---|---|---|---|
| 1 | 37.51 | 0.9392 | **191.59** | **57.76** |
| 2 | 37.58 | 0.9391 | 215.18 | 65.12 |
| 3 | **37.63** | **0.9403** | 238.76 | 72.79 |

Table 1 shows that as the number of iterations increases, PSNR and SSIM performance improves, but runtime also grows noticeably. To achieve a better balance between performance and efficiency, we ultimately choose 3 iterations as the default configuration.

## 3 VISUALIZATION OF HIDDEN FEATURES FROM THE EASA MODULE.

In this section, we visualize the intermediate features of the first EASA module to provide an intuitive illustration of how it leverages prior information to achieve effective representations. Figure 1 demonstrates that using gradient magnitude effectively preserves key information within the input features. By leveraging the selective attention mechanism, feature representations are further refined, leading to output features with enhanced structure and richer detail.

## 4 VISUALIZATION OF HIDDEN FEATURES FROM THE IFS BLOCK.

In this section, we visualize the intermediate features of the IFS block to intuitively demonstrate its process of dynamically merging valuable features from multi-exposure images. Figure 2 shows that

the two GRUs used with different kernel sizes, effectively fuse useful structural and detailed information from the multi-exposure features. By integrating these two complementary representations, the model achieves a more dedicated feature representation. Meanwhile, as the number of iterations increases, the model progressively suppresses irrelevant information and continuously enhances the learned features.

## 5 MORE VISUAL COMPARISONS

In this section, we provide additional visual comparisons between the proposed method and state-of-the-art approaches Dudhane et al. (2022; 2023); Kong et al. (2024); Liu et al. (2023); Zhang et al. (2025); Liu et al. (2022) on test datasets. Figures 3 and 4 present the comparison results, demonstrating that our method generates high-quality HDR images with more details.

## 6 VISUAL COMPARISON ON REAL-WORLD SCENES

In this section, we compare our method with TMRNet Zhang et al. (2025) and Joint-HDRDN Liu et al. (2023) on real-world scenes. As detailed in Section 3.1, these approaches rely on optical flow estimation and self-attention for inter-frame feature aggregation, which often fails under complex exposure conditions. Consequently, their reconstructions suffer from noticeable artifacts. In contrast, our proposed RawEnhancer consistently produces better results in these challenging scenarios, as shown in Figure 5.

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

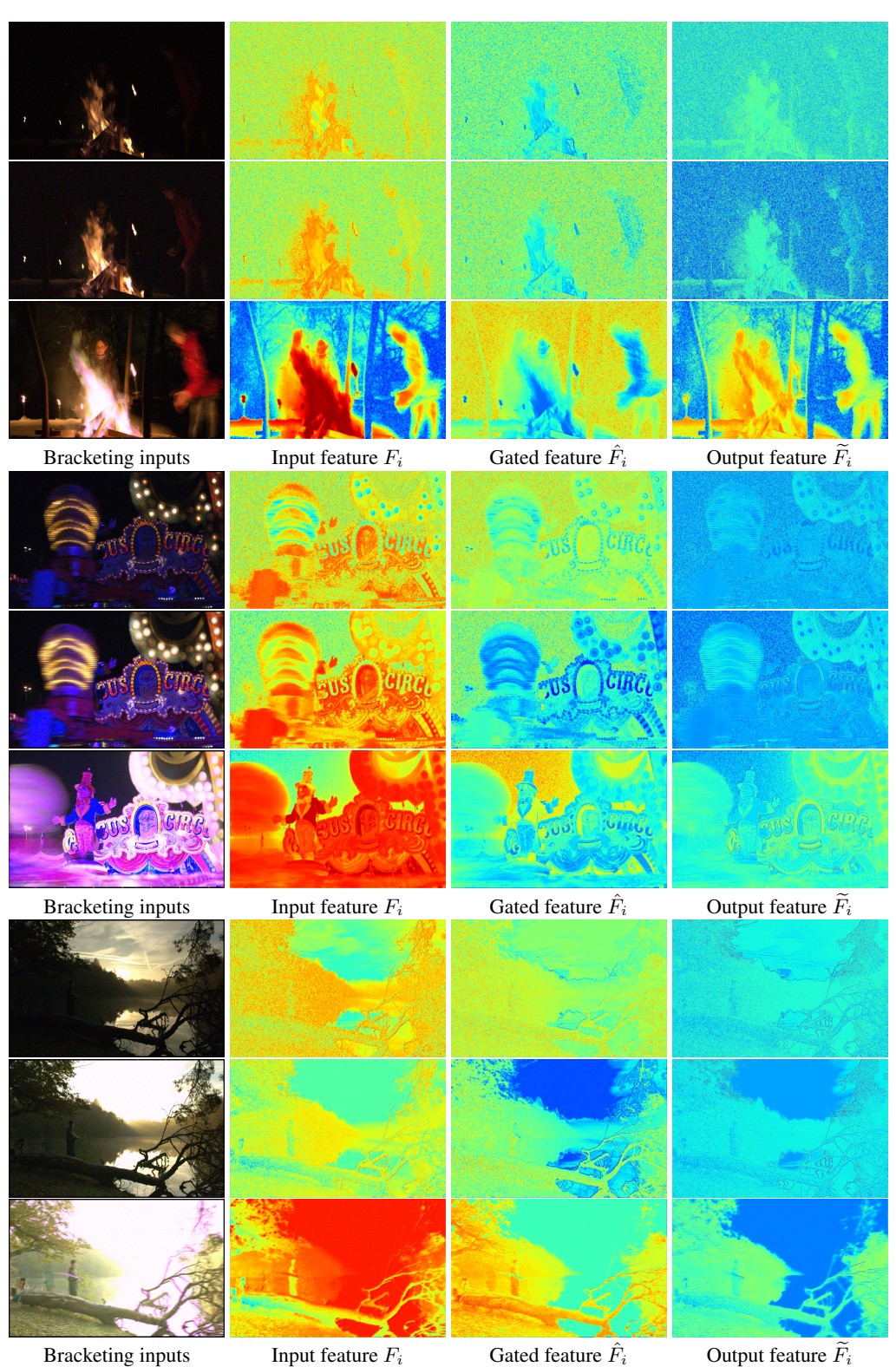

Figure 1: Illustration of the learned feature from the first EASA module. The EASA module can effectively utilize valuable information within images across various scenarios and exposures.

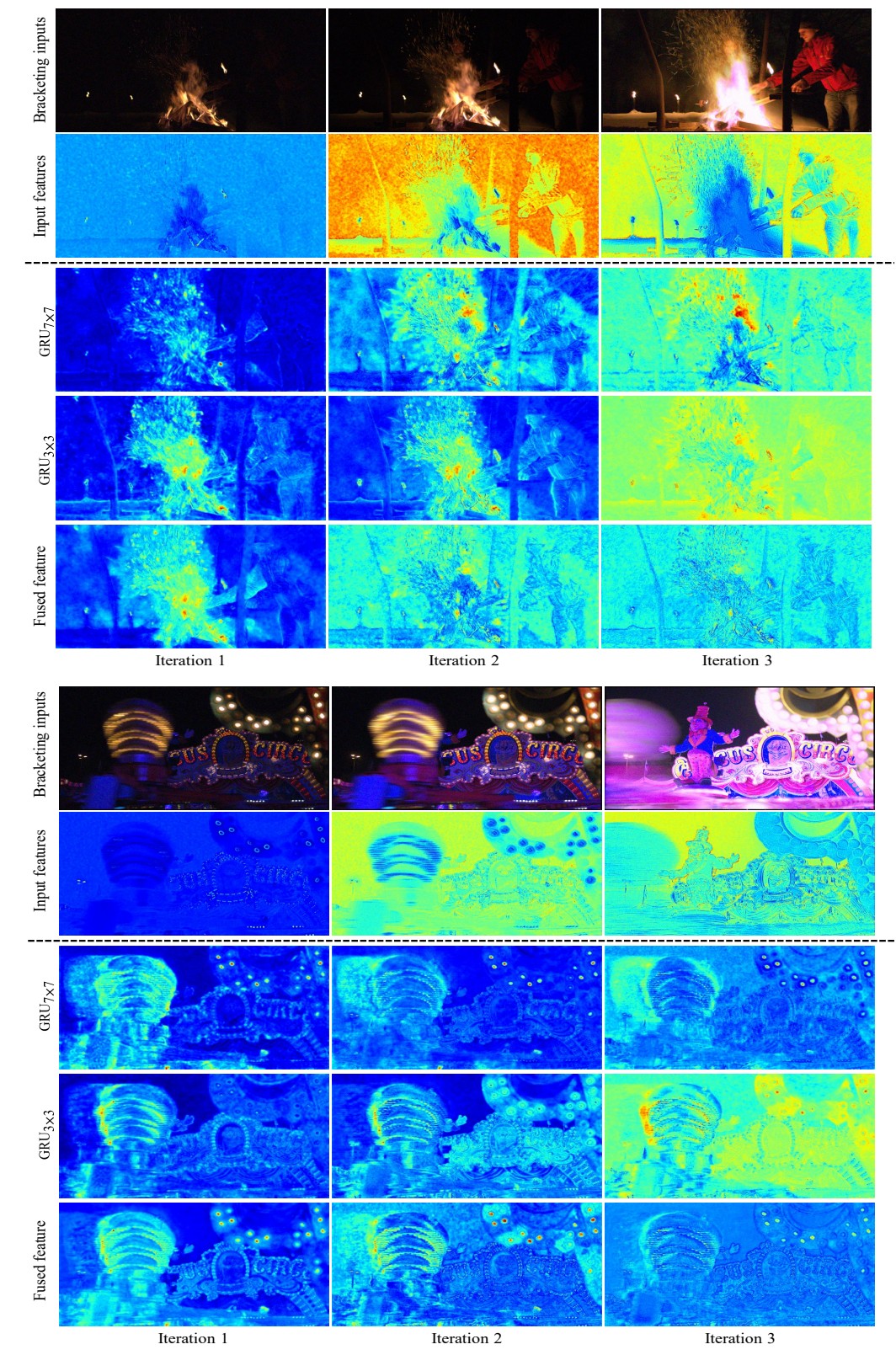

Figure 2: Illustration of the learned features from the IFS block at the $\times 2$ downsampled feature space. As the number of iterations increases, the model progressively suppresses irrelevant information and achieves a more discriminative representation.

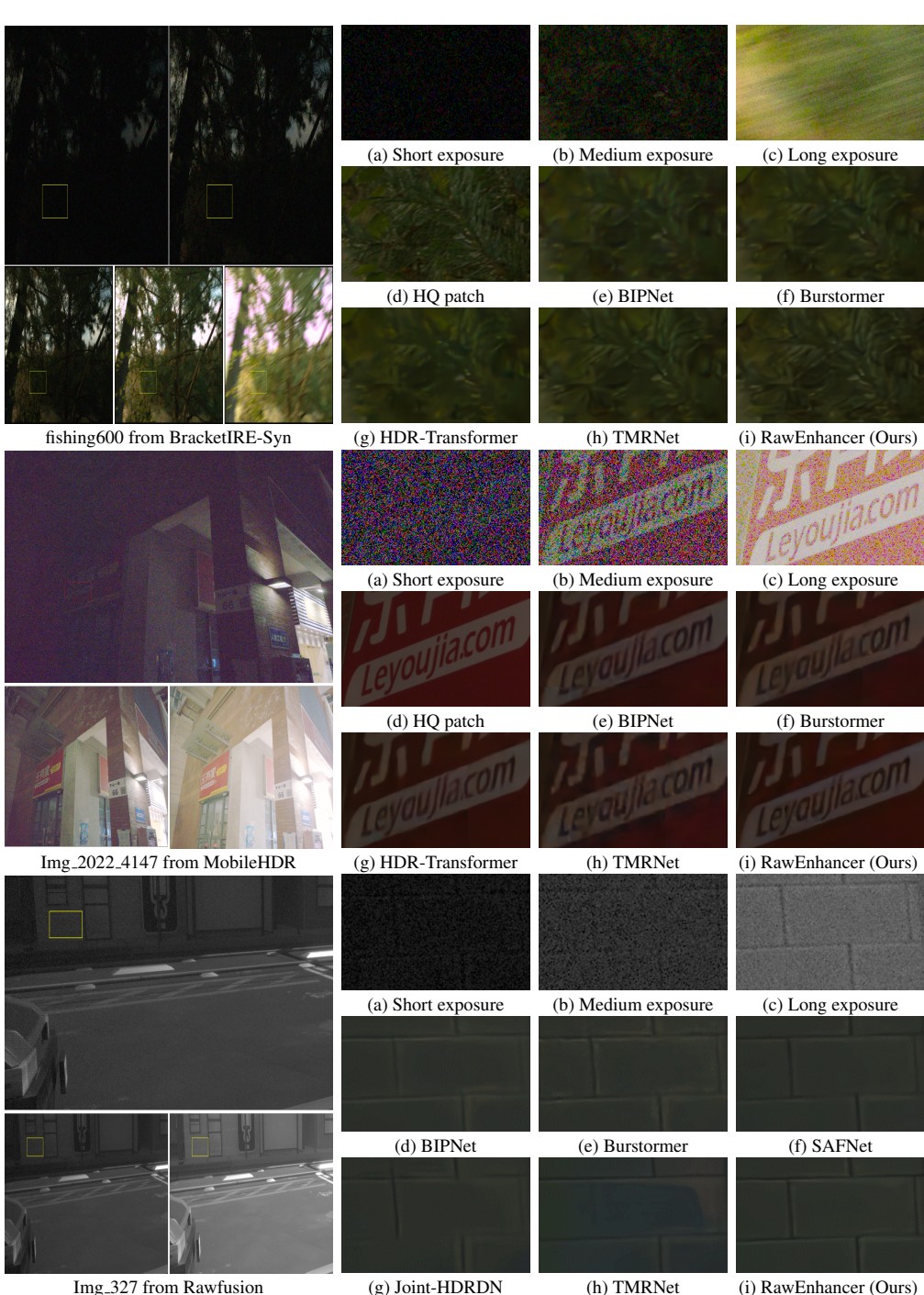

fishing600 from BracketIRE-Syn

(a) Short exposure   (b) Medium exposure   (c) Long exposure

(d) HQ patch   (e) BIPNet   (f) Burstormer

(g) HDR-Transformer   (h) TMRNet   (i) RawEnhancer (Ours)

Img_2022_4147 from MobileHDR

Img_327 from Rawfusion

Figure 3: Visual comparisons on different scenes. The reconstructed results of the listed methods exhibit noticeable artifacts, whereas our RawEnhancer produces sharper and cleaner images.

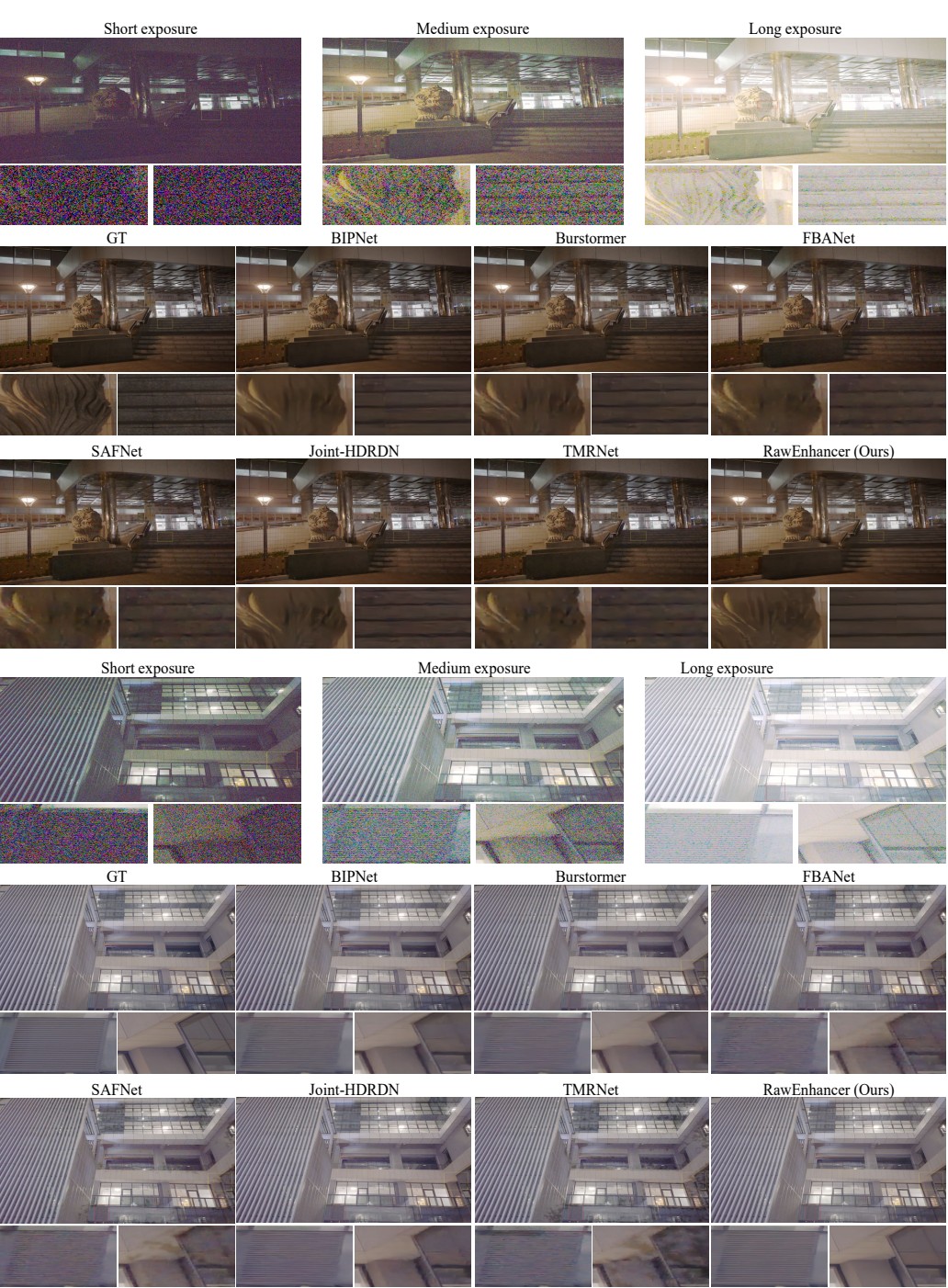

Figure 4: Visual comparison on MobileHDR Liu et al. (2023) dataset. The reconstructed results of the listed methods exhibit noticeable artifacts, whereas our RawEnhancer produces sharper and cleaner images.

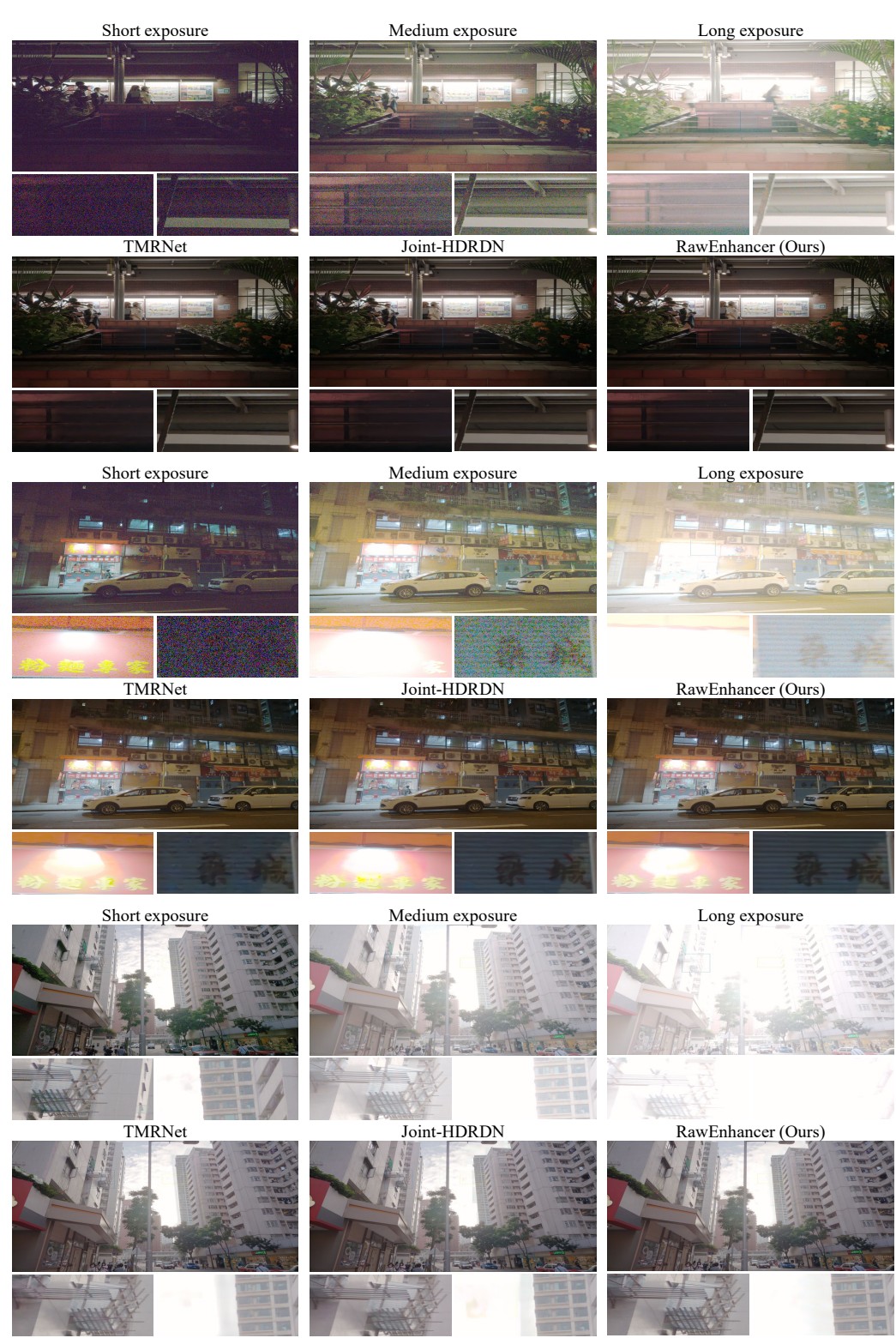

Figure 5: Visual comparison on real multi-exposure scenarios Liu et al. (2023). The reconstructed results of the listed methods exhibit noticeable artifacts, whereas our RawEnhancer consistently reconstructs images with better detail and structure across various complex exposures.