# OpenReview forum: "RawEnhancer: Attentive Multi-Exposure Selection for Bracketing Image Reconstruction"
_ICLR.cc/2026/Conference — ICLR 2026 Conference Withdrawn Submission_

### Official Review · Reviewer_9rSg · 2025-10-27

**Soundness:** 3
**Presentation:** 3
**Contribution:** 2
**Rating:** 4
**Confidence:** 3

**Summary:**

This paper proposes an alignment-free multi-exposure RAW HDR reconstruction framework.
The key ideas:
- Exposure-Aware Spatial Attention (EASA) to select useful regions per exposure.
- Inter-Feature Suppression (IFS) to avoid aggregating corrupted or saturated regions.
- Multi-branch architecture to exploit complementary exposures.

Experiments show competitive performance against strong multi-exposure methods such as TMRNet while being significantly more efficient in FLOPs and parameter size.
The problem is relevant and the practical advantages (no alignment, fast inference) are appealing.

**Strengths:**

- ligthweight compared to SOTA method (TMRNet)
   - no aligment stage, more robust to exposure/shift variation.
- clear motivation and architecture design
   - Exposure-aware selection is intuitive and well motivated.
- Competitive performance
   - Similar or better performance than previous methods on key benchmarks.

**Weaknesses:**

1. Limited task scope compared to recent SOTA
   - Only reconstruction (HDR + noise suppression)
   - No deblurring or super-resolution unlike TMRNet’s broader task coverage

2. Domain generalization not fully validated
   - Trained mostly on synthetic paired data
   - No self-supervised adaptation to real scenes

3. Exposure selection rationale could be better justified
    - Decisions rely on magnitude-based attention without theoretical illumination modeling

4. Real-world stress testing insufficient
    - Motion artifacts partially shown but more extreme cases needed
    - Limited camera diversity in datasets

5. Interpretability claims left mostly qualitative

6. Key claims are not validated
    - The paper continuously emphasizes that removing alignment improves robustness, but there are no experiments isolateds motion or exposure-shift scenarios, also there's no comparison against alignment-based versions.

7. Scientific contribution is not novel.
   - Exposure-aware spatial attention and feature suppression are incremental variations of existing techiques. There's only engineering imporvements on raw exposure correction.

**Questions:**

1. How robust is the model to significant motion and geometric misalignment?

2. Have you tested generalization on unseen cameras without fine-tuning?

3. Can the exposure-attention decision be explained via illumination or SNR models?

4. Could the model extend to complex tasks (denoise-deblur-HDR jointly) with small cost?

5. If the absence of alignment is a major claim, where is the empirical proof that optical-flow-based methods fail more often?

6. Does the model handle motion blur or non-rigid misalignment robustly?

7.  How well does it generalize to unseen camera sensors or ISP pipelines?

---

### Official Review · Reviewer_Rr5i · 2025-10-28

**Soundness:** 2
**Presentation:** 2
**Contribution:** 2
**Rating:** 2
**Confidence:** 4

**Summary:**

This paper proposes a network architecture called RawEnhancer for efficient bracketing image reconstruction to generate high-quality HDR images. The method employs an asymmetric encoder-decoder framework and introduces two key modules: an Exposure-Aware Selective Attention (EASA) layer and an Iterative Feature Selection (IFS) module. The EASA layer leverages gradient magnitude and color channel intensity priors from RAW sensor data to selectively encode multi-scale contextual information, identify valid regions, and suppress degraded content. The IFS module dynamically fuses complementary information from multiple exposures through progressive cross-scale feature propagation, eliminating the need for explicit feature alignment. Experimental results demonstrate that RawEnhancer performs favorably against state-of-the-art methods in terms of both accuracy and model efficiency, especially excelling in handling complex degradation scenarios.

**Strengths:**

1. The proposed RawEnhancer adopts an "encode-then-fuse" strategy, which differs from traditional methods that directly merge multi-exposure images at early stages. This approach more effectively leverages the inherent priors and complementary information within each RAW image.
2. The EASA layer utilizes gradient magnitude and color channel intensity priors to identify valid regions in images and suppress noise and saturation in overexposed or underexposed areas, thereby improving the ability to recover clipped pixels. This addresses the issue of traditional attention mechanisms being sensitive to luminance variations.
4. Superior Performance and Efficiency. Experimental results (as shown in Table 1 and Table 2) indicate that RawEnhancer achieves higher PSNR and SSIM values across multiple benchmark datasets. Simultaneously, it significantly outperforms many existing methods in terms of FLOPs, GPU memory consumption, and inference time, striking a good balance between performance and efficiency.

**Weaknesses:**

1. Lack of novelty. This paper features the architecture modification based on the UNet architecture. While, architecture modification is quite normal. This paper lacks theoretical introduction or insightful thinkings that deeply couple with the exposure fusion task.
2. The efficiency (Params, FLOPs，and GPU Mem) of this paper is inferior to the leading medels, for example, HDR-Transformer. This also raises another issue that the performance gain may stem from the larger model size instead of the modified architecture.

**Questions:**

Please refer to the weakness part.

---

### Official Review · Reviewer_fuzB · 2025-10-30

**Soundness:** 2
**Presentation:** 3
**Contribution:** 2
**Rating:** 4
**Confidence:** 4

**Summary:**

This paper presents RawEnhancer, a novel network for bracketing image reconstruction. The authors introduce an exposure-aware selective attention mechanism and an iterative feature selection module for effective feature encoding and fusion. Experimental results show that RawEnhancer achieves competitive performance compared to state-of-the-art methods.

**Strengths:**

1. The overall performance is satisfactory, and the proposed method achieves a reasonable balance between accuracy and efficiency.

2. The research topic is meaningful and has broad potential applications.

3. The paper is clearly written and well-organized.

**Weaknesses:**

1. The novelty of this work is somewhat limited. As illustrated in the manuscript, the proposed model still follows a U-shaped architecture. Although several modifications are introduced to some basic modules, these changes mainly combine existing techniques from previous works rather than presenting fundamentally new ideas.

2. The paper lacks convincing evidence to support its main motivation. The authors claim that the two-stage framework is suboptimal under large exposure variations; however, no clear justification is provided for why the proposed network is more robust to such exposure differences. In fact, the proposed method appears quite similar to existing attention-based approaches and may suffer from the same limitations.

3. The description in Section 3.1 is confusing. From Eq. (1) and Eq. (2), it seems that features are extracted separately for each exposure, whereas Figure 1 suggests that feature extraction is performed jointly. The authors should clarify this inconsistency and improve the overall explanation.

4. The visual results presented in the ablation study do not sufficiently demonstrate the effectiveness of the proposed gradient magnitude gating mechanism. More quantitative or qualitative evidence is needed to support this claim.

**Questions:**

1. In lines 366–367, the authors state that SAFNet and Joint-HDRDN are excluded because they only support three-exposure inputs. However, HDR-Transformer also accepts only three exposures as input. Could the authors clarify why HDR-Transformer can still be directly applied to the BracketIRE-syn dataset?

---

### Official Review · Reviewer_oeZH · 2025-11-01

**Soundness:** 2
**Presentation:** 2
**Contribution:** 2
**Rating:** 2
**Confidence:** 4

**Summary:**

This paper proposes RawEnhancer, an asymmetric encoder-decoder network for high-dynamic-range (HDR) reconstruction from bracketed RAW image sequences. The authors identify a key failure in prior work: alignment-based fusion (using optical flow or attention) fails in over/underexposed regions where luminance consistency is lost. To solve this, RawEnhancer adopts an "encoding-then-fusion" strategy. First, a weight-sharing encoder processes each RAW exposure individually to extract multi-scale features. This encoder is built on a novel Exposure-Aware Selective Attention (EASA) block, which uses the gradient magnitude of the RAW input as a spatial gate to select reliable, content-rich features and suppress noise or saturation before applying channel-wise attention. Second, the multi-scale features from all exposures are fed into the decoder, which uses an Iterative Feature Selection (IFS) module to progressively fuse them without explicit alignment. The IFS block is reference-free and uses parallel GRUs with different kernel sizes ($3 \times 3$ and $7 \times 7$) to model both local and non-local spatial dependencies during fusion. Experiments on BracketIRE-syn, MobileHDR, and RawFusion datasets show the method is SOTA-competitive in quality while being significantly more efficient (faster runtime, lower FLOPs/memory) than prior methods.

**Strengths:**

1. The paper correctly identifies a valid and practical failure mode in prior alignment-based HDR methods, where alignment mechanisms break down in saturated or noisy regions.
2. The proposed "encoding-then-fusion" architecture is explicitly designed to be lightweight and efficient (Table 2), which is a valuable engineering goal for a practical task like HDR reconstruction.

**Weaknesses:**

1. The experimental comparison is weak and misses the most relevant recent work. The baselines cited are largely from 2023 or earlier. The paper fails to compare against any recent SOTA methods (e.g., [1, 2, 3, 4] etc.), which have significantly advanced the field.
2. The paper's architecture (a CNN-Transformer hybrid) is already somewhat dated. There is no discussion or comparison to recent State-Space Models (e.g., Mamba), which are highly relevant for sequence-based tasks like fusing an sequence of exposure frames and are showing dominant performance in vision.
3. The core components are derivative of existing work. The EASA block is a minor variant of the gated attention mechanisms in NAFNet and Restormer, simply replacing the internally-learned gate with an externally-computed gradient. The IFS module uses standard GRUs, a well-known technique for fusion, and its performance is heavily dependent on the NAFNet blocks it contains.

[1] Exposure Bracketing Is All You Need For A High-Quality Image, ICLR 2025
[2] UltraFusion: Ultra High Dynamic Imaging using Exposure Fusion, CVPR 2025
[3] Neural Exposure Fusion for High-Dynamic Range Object Detection, CVPR 2024
[4] NTIRE 2024 Challenge on Bracketing Image Restoration and Enhancement, CVPRW 2024
[5] MobileMEF: Fast and Efficient Method for Multi-Exposure Fusion
[6] ExpoMamba: Exploiting Frequency SSM Blocks for Efficient and Effective Image Enhancement

**Questions:**

1. Can the authors justify the omission of all recent SOTA HDR fusion methods from [1-6]. Without this, the performance claims are unverified.

2. Given the sequence-to-sequence nature of fusing $t$ exposure frames, have the authors considered benchmarking against recent State-Space Models (e.g., Mamba, as in [6])? How do they expect their GRU-based fusion to compare in terms of both performance and efficiency?

3. The IFS module's performance seems heavily reliant on the NAFNet blocks $N_{f}(\cdot)$  for refinement. Can the authors provide an ablation study that isolates the contribution of the dual-GRU fusion, for example, by replacing $N_{f}(\cdot)$  with a simple MLP or removing it entirely?

4. The IFS module is described as running for a fixed 3 iterations, but the model accepts a variable number of input frames ($t$). How are the $t$ frames from the set $S_i$ processed by the GRU? Are they concatenated? Processed sequentially? Please clarify this ambiguous mechanism.

---

### Note · Authors · 2025-11-19

I have read and agree with the venue's withdrawal policy on behalf of myself and my co-authors.